# Autophase V2: Towards Function Level Phase Ordering Optimization

Mohammed Almakki[§]
*University of Khartoum*
mohammed.a.h.almakki@gmail.com

Ayman Izzeldin[§]
*University of Khartoum*
ayman.izzeldin@gmail.com

Qijing Huang
*NVIDIA*
jennyhuang@nvidia.com

Ameer Haj Ali
*Anyscale*
ameer@anyscale.com

Chris Cummins
*Facebook*
cummins@fb.com

*Abstract*—Compilers are equipped with optimization passes that can be applied to improve the quality of a program. The selection and ordering of these passes is a classic NP-hard problem known as the phase ordering problem. Traditionally, compilers use expert-picked sequences to optimize for performance (e.g. -O2, -O3), or code size (e.g. -Os, -Oz). However, not all programs respond positively to all optimizations, and prior work has shown that these expert-picked phase orders can be outperformed by tailoring the phase order decisions to individual programs.

In this work, we propose further specializing the phase ordering for each function. We investigate the impact of various function and module passes and show that function-specific phase ordering provides an extra 2.3% improvement in code size reduction compared to program-specific phase ordering using a cheap search budget. Compared to using the Oz flag, the deep reinforcement learning method achieves up to 9% more code size reduction in the same dataset, and up to 6% improvement when transferring to new unseen data. Our exhaustive search experiment shows that searching on different levels of abstraction will be beneficial for solving the phase ordering problem. However, we note several limitations with our reinforcement learning approach, the observation features are not sufficient to give better generalization, the reward has high variance and the datasets are not well representative for all programs.

## I. INTRODUCTION

Code optimization is an important step in developing applications. Many applications require critical constraints for code performance. For example in applications that targets low resource hardware the size of the code should be as minimum as possible. Achieving a small code size reduction will be beneficial for these kind of applications. In addition, other factors like the compiler-time and run-time are important for some applications. Applications developers use the tools provided by the compilers to optimize their code. These tools require manual work and are time consuming. Integrating intelligent systems with compilers that automate the process of code optimization will improve the process of developing applications and help the Applications developers with their work.

The optimizations performed by a compiler vary in their purpose and results. Most importantly, these optimizations are applied in a sequence, one after the other, and interact and affect the output of each other in ways that affect the output of a compiler. That is to say, the order in which these optimizations are applied affects the target program in ways that can have either a negative or positive impact on its performance. Further, not all optimizations are suitable for all programs. Choosing a specific order for these optimizations to achieve the best outcome based on some performance metric is known as the phase-ordering problem [16].

The current compilers either use predefined orders of passes controlled by command-line flags, to achieve different optimization levels that may or may not be optimal for the program or allow the programmer to manually specify a sequence of passes to apply to the program. Generally, the hard-coded order is good but is not optimal for every program, as one fixed optimization sequence is highly unlikely to be the optimal sequence for every program.

Recently, deep learning methods, especially deep reinforcement learning, have been used to solve optimization problems. In the context of the phase-ordering problem, several authors proposed methods to try and solve this problem using deep reinforcement learning [6], [7], [14]. These methods showed that different programs benefit from having different optimization pipelines. Reinforcement learning has been used to make agents learn to apply different optimization ordering for different modules such that it produces better results compared to the static sequence used in compilers nowadays.

However, work on this problem so far has not experimented with applying different optimization phase orderings to different segments of a program. Our goal is to explore this possibility and observe its effect on the overall program performance. Towards this goal, we leverage the infrastructure provided by LLVM and CompilerGym [4] to enable the application of different optimization phase orderings on different functions. We investigate searching different optimization orderings for different functions in the program. To demonstrate our method we use two approaches, random search and deep reinforcement learning. Our results show that our method is promising. In this paper, we make the following contributions:

- We propose a novel technique for phase ordering problem

---

[§]Equal contribution

that searches for optimization sequences on the function level, rather than the whole-program level.

- We show that using a cheap time budget doing function-specific phase orderings gives a 2.3% improvement in code size reduction over program-specific phase orderings using random search.
- We implement a deep reinforcement learning framework for function-specific phase ordering problem based on CompilerGym [4] environments. We train a policy on different datasets and show that it outperforms the $-Oz$ flag by 1.5% in the cBench dataset and 9% in CHStone [8] dataset. Our method shows a good generalization by achieving better or similar performance compared to the $-Oz$ flag in two different unseen datasets.

## II. MOTIVATION

To demonstrate the need for per-function phase ordering we performed an exhaustive search experiment on *dijkstra* module from the cBench dataset which consists of 5 functions[1]. For every function, using 13 function passes[2] and a sequence length of 5 passes, we searched for the optimal sequence that gives the most code size reduction by enumerating all possible sequences. The same experiment is done on the module level, i.e. the same sequence of passes is applied to every function in the module. Comparing the results of the two experiments, the function level method outperforms the module level by 2% in code size reduction shown in Table II. Table I shows the optimal sequences obtained from the experiment. This clearly shows that selecting different sequences of function passes for each function yields more reduction than applying a single sequence for all of the functions.

#### TABLE I
SAMPLE OF OPTIMAL PASS SEQUENCES FROM THE EXHAUSTIVE SEARCH EXPERIMENT

| Function | Shortest Optimal Sequence Sample |
| --- | --- |
| print_path | -gvn -instcombine |
| enqueue | -mem2reg -early-cse -simplifycfg |
| dequeue | -sroa -early-cse |
| dijkstra | -gvn -instcombine -reassociate -simplifycfg |
| main | -gvn -instcombine -simplifycfg |

| Module | Shortest Optimal Sequence Sample |
| --- | --- |
| dijkstra | -gvn -instcombine -reassociate -simplifycfg |

## III. METHODOLOGY

This section describes our proposed approach for per-function phase ordering.

[1]We omit the function *qcount* as it only consists of 2 instructions and is not optimizable with the experiment setup

[2]The passes used for the exhaustive search experiments are: *gvn, loop-reduce, loop-deletion, reassociate, loop-rotate, early-cse, adce, instcombine, simplifycfg, dse, loop-unroll, mem2reg, sroa*.

#### TABLE II
THE INSTRUCTION COUNT REDUCTION OF THE OPTIMAL PASS SEQUENCES FROM THE EXHAUSTIVE SEARCH EXPERIMENT

| Method | Module code size reduction compared to $-Oz$ |
| --- | --- |
| Module Level | 0.964x |
| Function Level | **0.984x** |

*1) Overview:* In LLVM, every program consists of a number of translation units called modules written in a platform-independent intermediate representation (IR). Each module contains, among other things, one or more functions which themselves contain a list of basic blocks. A basic block consists of a list of instructions. The optimization passes in LLVM are divided into different types depending on how those passes work and what code segments they target. Two of these types are the function passes and module passes. As the names suggest, the function passes apply an optimization on a single function in a module, whereas module passes apply an optimization on the whole module.

In the current implementation of LLVM, the opt tool is used for managing and applying passes on modules. When applying a function pass using opt tool on a module the function pass will be applied for every function in the module. This approach is not efficient as applying the function pass will not be optimal for all functions. Moreover, there is a high probability that the action will not affect the function as the previous work [9] shows that the probability of a pass in $-O3$ flag pipeline to modify the input code is very low.

Given that, we introduce a new technique for solving the phase ordering problem. Our method solves the phase ordering problem by finding the best optimization sequences for every function in the module separately. Figure 1 shows the difference between opt tool and our method.

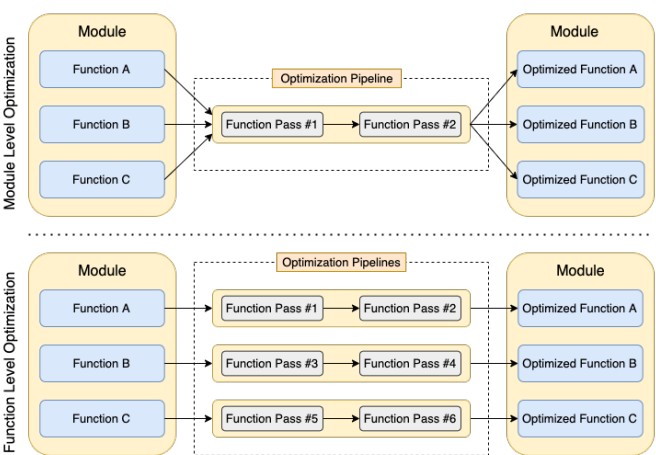

Fig. 1. Comparison of the current opt tool method of applying the same function pass on every function and our method of using different phase orderings for different functions.

*2) Deep Reinforcement Learning Environment Formulation:* Reinforcement learning is a sub-field of machine learning. It

has a different learning paradigm than common supervised machine learning methods. It models the problem as an agent that acts on a constrained environment with a state and a mechanism to reward the agent. Reinforcement learning problems involve learning that maps situations to actions in order to to maximize a numerical reward signal. The goal of reinforcement learning is to maximise this reward. Deep reinforcement learning utilizes deep learning in finding the solution to the reinforcement learning problem.

Let $N$ be the number of steps in the environment. At each step the agent can apply one of $K$ transformation function passes. The passes are applied on one function which represents the state $s$ of our environment. The state is not fully observable and is modeled by a number of $M$ features that represents the observation $o$ of the environment. The agent selects one function pass in every step and tries to minimize the code size reduction. The spaces in our environment are defined as:

**Action Space**: It consists of $K$ transformation function passes. Then action $a$ is defined as $\{a : a \in [0, K)\}$.

**Observation Space**: The observation $o$ is an $M + K$ vector constructed by concatenating two vectors $o_f$ and $o_h$. $o_f$ is $M$ dimensional function feature vector. $o_h$ is $K$ dimensional action history histogram vector. At each step the entry associated with the applied action $a$ is increased by one.

**Reward**: The reward $R(s_t)$ of transitioning from state $s_t$ to $s_{t+1}$ is defined as:

$$R(s_t) = \frac{C(s_{t-1}) - C(s_t)}{O0(s_{t=0}) - Oz(s_{t=0})} \quad (1)$$

Where $t$ is the timestep t. $s_t$ is the observation at timestep t. $C(s_t)$ is the IR instruction count of the function at time t. $O0(s_{t=0})$ is the IR instruction count of the function at $t = 0$ optimized using $-O0$ flag. $Oz(s_{t=0})$ is the IR instruction count of the function at $t = 0$ optimized using $-Oz$ flag.

*3) Environment Implementation:* To create our environment, we modified the LLVM environment in CompilerGym [4] library to allow working on specific functions within the module. For this, we introduced function benchmarks, benchmarks that allow running function passes on specific functions in the module. Additionally, we added multiple observations to the environment for the function-specific information.

## IV. EXPERIMENTAL SETUP

In this section, we give a general overview of how we set up the random search and reinforcement learning experiments.

### A. Random Search Experiments

To evaluate our method we first start by doing random search experiments. The random search is carried out on every function on the module. The search has two parameters, time ($t$) and patience ($p$). Those parameters are defined as follows:

- Time: It is the search time for a specific function. The search time is fixed. For example, if $t = 10$, then the search will run for 10 seconds for each function.

- Patience: The patience of the search is the number of steps allowed without improvement. For example, if $p = 30$, the search will stop after 30 steps are ran without any improvement in the reward. The patience ratio is the ratio of patience to the number of actions in the action space.

The random search on a single function proceeds by selecting random actions from the action space to evaluate. The random search is restarted if the patience condition is not met and stopped if the search time has elapsed. These experiments were based on CompilerGym [4] random search experiments using the cBench dataset modules. We add support for our function environment to the random search experiments code in CompilerGym [4] by splitting each module into its functions and then running the search on each function with the function environment. After the search is finished the final function pipelines are applied for every function and the whole module instruction count reduction is calculated.

### B. Reinforcement Learning Experiments

We use the Ray RLlib [11] framework for running reinforcement learning experiments with the proximal policy optimization (PPO) [15]. The RLlib [11] framework is an open-source distributed reinforcement learning framework. Moreover, we use Tune [12], a hyperparameter tuning framework, to run our experiments.

## V. EXPERIMENTAL RESULTS

### A. Random Search Experiments

We ran random search experiments on 15 benchmarks selected from cBench dataset. The benchmarks are selected only if they include less than 150 functions, which amounts to 433 functions from the selected benchmarks. Three experiments were conducted, two on the module level and one on the function level. The three experiments' setup is as follows.

- Function level experiment: We use all of the 91 function passes available in the CompilerGym [4] LLVM environment as the action space. The experiment is carried out on the functions of the 15 benchmarks. The search time is 120 seconds per function and the patience length is 113. The patience is calculated based on the number of function passes (91) times the patience ratio (1.25).
- Module level experiments: We use all the available 124 passes on CompilerGym [4] LLVM environment as the action space. The experiments are carried out on 15 modules. The search time for one experiment is 3600 seconds per module and 120 seconds per module for the other. Both experiments use a patience length of 155. The patience is calculated based on the total number of passes (124) times the patience ratio (1.25).

Figure 2 shows the code size reduction for each benchmark from the three experiments. To compare these experiments we calculated the geometric mean of the code size reduction for the 15 modules. Table III summarizes the results of the three experiments.The results show that the function level phase

ordering experiment outperforms the module level phase ordering experiment using less search time and shorter patience length.

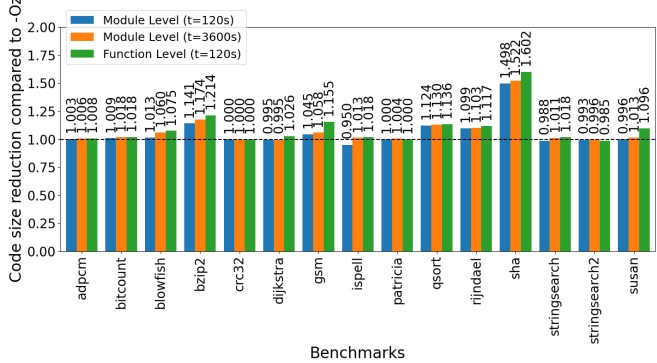

Fig. 2. Code size reduction comparison for the random search experiments on 15 benchmarks from the cBench dataset.

TABLE III
RANDOM SEARCH EXPERIMENTS RESULTS

| Method | Time (s) | Patience | Geomean module code size reduction compared to $-Oz$ |
|---|---|---|---|
| Module level | 3600 | 155 | 1.066x |
| Module level | 120 | 155 | 1.050x |
| Function level | 120 | 113 | **1.089x** |

## B. Deep Reinforcement Learning Experiments

*1) Deep Reinforcement Learning Performance:* We trained two RL agents using our method. For every dataset, we split the modules into functions, then with every new episode a new function is loaded in the environment. After all functions are done the environment will go back to the first function in the dataset. In addition, we removed every function that is removed by $-Oz$ as it is trivially optimizable. In both experiments, we used an episode length of 45 and trained for $50K$ episodes using the PPO algorithm with the default configuration. Training and testing was done on the same datasets. After training, the policy is rolled out only once for every function in the test benchmarks for the same number of steps. Figures 3 and 4 show the results of the two experiments on cBench[3] and CHStone datasets respectively. The results show that our method gives better results than the $-Oz$ flag. Also, in $sha$ in Figure 3 and $adpcm$ in Figure 4 the agent was able to achieve approximately 40% improvement over $-Oz$ flag.

*2) Reinforcement Learning Generalization:* To test the generalization of our method. We trained an agent on 100K functions from AnghaBench [5] datasets. We used an episode length of 45 and trained for $100K$ episodes using PPO [15] algorithm with the default configuration. Then we tested the agent on three unseen datasets (Csmith, cBench[3] and CHStone) by rolling out the trained policy only once for every

[3]Excluding $ghostscript$ benchmark

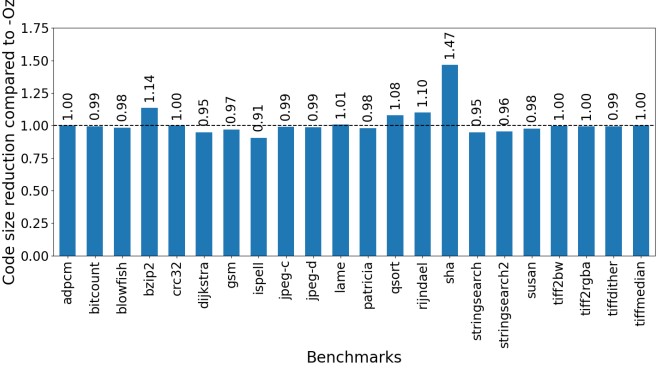

Fig. 3. Code size reduction results for the RL agent trained and tested on cBench dataset[3]. The geomean module code size reduction relative to $-Oz$ is 1.015x.

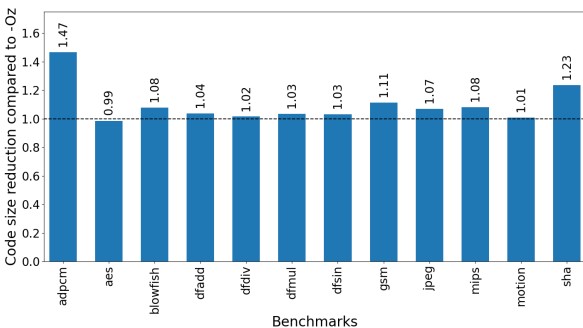

Fig. 4. Code size reduction results for the RL agent trained and tested on CHStone dataset. The geomean module code size reduction relative to $-Oz$ is 1.090x.

function in every benchmark in the test dataset for the same number of steps. The results are shown in table IV. From the results, it shows that the agent can get positive results over $-Oz$ flag on the CHStone dataset and near performance to $-Oz$ on the cBench dataset.

TABLE IV
TEST DATASETS RESULTS

| Dataset | Geomean module code size reduction compared to $-Oz$ |
|---|---|
| cBench | 0.999x |
| CHStone | 1.064x |
| Csmith | 0.945x |

## VI. RELATED WORK

Different approaches have been proposed to tackle the phase ordering problem. Kulkarni et. al [10], used an evolutionary-based method for selecting the order of optimization passes. MiCOMP [1] used neuro-evolution to construct an artificial neural that predicts good optimization passes. Cereda et. al [2], used a collaborative filtering approach from the field of recommendation systems in compiler optimization. ICMC [13] combines metric learning and collaborative filter to find better optimization orderings. BOCA [3] uses Bayesian optimization

to tackle the phase ordering problem. Jayatilaka et. al [9] used two machine learning approaches to reduce the compilation time.

Reinforcement learning-based methods are also used for tackling the phase ordering problem. The Autophase [7] framework applies deep reinforcement learning to the Autophase problem. The framework achieves 28 precent circuit speedup over the O3 compiler flag. CORL [14] framework Also uses deep reinforcement learning to optimize the order of optimization passes. The framework was able to achieve 1.32x speedup on previously-unseen programs. CompilerGym [4] is a multi-task reinforcement learning environment for compiler optimization. A trained agent using this environment achieves positive code size reduction results over the Oz compiler flag.

## VII. CONCLUSION AND FUTURE WORK

We present a new technique for tackling the phase ordering problem by applying different phase orderings to different functions inside a single module. We show that our approach gives better code size reduction using random search with a cheaper search budget compared to module-level search for an optimal optimization sequence. We also trained two deep reinforcement learning agents on function benchmarks and were able to get an improvement over $-Oz$ performance. Several limitations however need to be acknowledged. Our current approach for the reinforcement learning agent uses the Autophase [7] features vector for the observation space, which hinders the agent's ability to generalize due to the sparsity of these features when extracted from a single function. Further research is needed to identify a set of features that work best for functions. In addition, there is high variance in the reward during training because the functions are diverse, some functions are easily optimized and some functions are not. The high variance in reward affects the performance of the reinforcement learning agent. As such, a good dataset that represents a lot of functions is needed to achieve better generalization. Further, these experiments focused on using function passes on the function level, and completely omitted the use of module passes. Further research should be done to investigate the addition of module passes within the optimization pipeline.

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
