# OpenReview forum: "Autophase V2: Towards Function Level Phase Ordering Optimization"
_iscaconf.org/ISCA/2022/Workshop/MLArchSys — MLArchSys 2022_

### Official Review · Reviewer_R2M5 · 2022-05-18
**Autophase V2: Towards Function Level Phase Ordering Optimization**

**Rating:** 7
**Confidence:** 4

**Review:**

The paper proposes a search based approach for the compiler phase ordering problem.

Strengths:
* Interesting idea, paper is well written and concise in conveying the approach & results.
* Provides a finer grain function level pass ordering optimization using random search + RL agent and shows that a relatively cheap search for pass ordering can outperform Oz in code size reduction.
* Demonstrates RL agent trained+tested on the same dataset can find favorable pass orderings over Oz.
* Demonstrates that the RL agent trained on a held out dataset can somewhat generalize to unseen dataset, although limited.

Weakness:
* Provides relatively incremental improvement.
* RL cannot reach the random search improvements.
* Does not mention any improvement for the search time using RL over random search.
* Evaluation is limited as highlighted in the future work.
* RL agent generalization to unseen tasks are relatively poor (1 out of 3 tasks show improvement over Oz)

---

### Official Review · Reviewer_kdR6 · 2022-05-18
**Interesting idea, but limited experiments. Adding more insights about the optimization outcomes makes the paper more interesting**

**Rating:** 7
**Confidence:** 4

**Review:**

The paper explores an interesting venue for compiler optimization, applying phase ordering optimization at different granularity. More specifically, the authors suggest that applying the phase ordering optimization at the function-level granularity provides marginally better improvement in terms of code size compared to an optimization at the granularity of a module. Overall, I like the idea but the experimental setup is very limited (which seems to be sufficient for a workshop venue).

A couple of suggestions:

- It is not clear whether the `generalization` approach is a one-shot setting or it still performs a limited search in the space.
- It would be good if the authors can add some intuitions about when a function-level optimization could be beneficial.
- The function-level optimization adds more overhead to the search process. For a head-to-head comparison between different approaches, it may be better to perform both function-level and module-level optimization with the same search time.
- While I understand that it is hard to provide insights about the similarity between train and test benchmarks, but I am curious to see if there are some patterns between the final optimization pattern between train and test benchmarks. It may be interesting to provide a more detailed study of the final patterns to better understand the capability of the proposed approach in identifying generalized solutions.
- (not expert) but would it be useful to have both function-level and module-level phase ordering as a hierarchical optimization. It seems this could be an interesting venue to explore.

---

### Official Review · Reviewer_Uid1 · 2022-05-20
**Autophase V2: Towards Function Level Phase Ordering Optimization**

**Rating:** 6
**Confidence:** 2

**Review:**

The paper proposes function level phase ordering optimization in the compiler pass and reports reduction in code size as compared to the -Oz flag. The paper uses benchmarks and datasets from cBench, chstone and smith dataset. Specifically, the paper uses reinforcement learning and random search based phase ordering optimizations and claims that function level phase optimization uses lesser search time as compared to module level phase optimization.

Significance of the work:
- With complex functions being used in algorithms today, it is indeed imperative to have function specific optimizations rather than a single optimization which might not work well for all the functions. Hence, the problem the paper tries to address is important.

- However, on the other hand, end-to-end applications are build using various types of complex functions. However, these applications are mostly designed using a series of inter-connected microservices. Each Microservices can be compiled independently and packaged into a container. The Microservices approach indirectly uses function level optimization since each Microservices implements a single function. It would be interesting to see how this work might benefit the microservices architecture of building large application.

Evaluation:
- The evaluation could be more thorough and especially interesting when all other passes/optimizations of the compiler is run on the benchmarks and the final result is reported. How does for instance how does the code size reduction change when a function level pass is followed by a module level pass?
- The analysis could be more thorough. It was difficult to understand the reasoning behind a particular benchmark had higher reduction in code size for function level phase optimization as compared to module level.

---

### Decision · Program_Chairs · 2022-05-30

**Decision:**

Accept

**Comment:**

All the reviewers found the idea presented in this paper interesting and timely.  However, there are some concerns about evaluation setup and head-to-head comparison. It would be great if the authors can provide more details about the evaluation setup, particularly `the choice of benchmarks`, and `search time comparison between random and RL`. More importantly, one reviewer described that the results from RL is on part with random search. While this is an interesting application of RL, it would be great if the authors can provide some insights about the limitation of this approach, especially compared to a seemingly cheap random search algorithm. Also, one reviewer raised the concern about `generalization`, it was not clear how similar the source and target applications are.

Because of the positive reviews and the ratings, the paper was nominated for `Best Paper Award`.